# Interferon Signaling in the Endometrium and in Endometriosis

**DOI:** 10.3390/biom12111554

**Published:** 2022-10-25

**Authors:** Yuri Park, Sang Jun Han

**Affiliations:** 1Department of Molecular and Cellular Biology, Baylor College of Medicine, Houston, TX 77030, USA; 2Dan L. Duncan Cancer Center, Baylor College of Medicine, Houston, TX 77030, USA; 3Reproductive Medicine, Baylor College of Medicine, Houston, TX 77030, USA

**Keywords:** endometriosis, endometrium, interferon, cytokine, inflammatory disease

## Abstract

Endometriosis is an estrogen-dependent inflammatory disease that develops in reproductive-aged women who experience pelvic pain and infertility. Even though endometriosis is not a new disease, its molecular etiology has not been clearly elucidated. Defects in the immune system might be one of the factors that promote endometriosis progression. For example, elevated levels of proinflammatory cytokines are associated with endometriosis. Interferon is one of the cytokines that is elevated in endometriotic tissues compared with normal endometrium. Therefore, high interferon levels play a crucial role in endometriosis progression. In addition to endometriosis, however, interferon has a critical role in endometrial function, particularly in the initiation and maintenance of pregnancy. Therefore, this review describes the double-edged sword of interferon signaling in normal endometrial function versus endometriosis progression and also discusses interferon targeting as a new nonhormonal therapy for endometriosis. This approach may increase the efficacy of endometriosis treatment and reduce the adverse effects associated with current hormonal therapy for this disease.

## 1. Introduction

Endometriosis is the growth of endometrial lesions outside the uterus, such as in the ovaries and peritoneal cavities [1]. Three different phenotypes of endometriosis are found in the pelvic cavity: superficial peritoneal endometriosis (SUP), ovarian endometrioma (OMA), and deeply infiltrating endometriosis (DIE) [2,3]. Red lesions are found on the surface of the peritoneum in SUP, while endometriotic tissues are invaginated into the ovary in OMA. In DIE, endometriotic tissue invades other organs, including the cervix and rectum [2]. Endometriotic lesions that are shown in red are early, active lesions with vascularization. The lesions become black after inflammatory reaction and scarification and eventually turn white [2]. Up to 5–10 percent of women of reproductive age experience symptoms of endometriosis [1]. Symptoms include mild to severe pelvic pain, infertility, painful urination or bowel movements, and abnormal menstrual flow. Surgical resection of the lesions is the primary treatment for endometriosis, but surgery cannot prevent disease relapse [4]. The 2-year and 5-year post-operative recurrence rates are about 21.5 percent and 40–50 percent, respectively [5]. The median time of post-operative recurrence is about 30 months [6]. Additionally, endometriosis is an estrogen-dependent disease [1,7]. Therefore, hormone-suppressive drugs that block the synthesis or activity of estrogens (E_2_), such as elagolix [8], progestin [9], and Danazol [10], have been applied to relieve endometriosis symptoms. Hormonal suppression treatment after conservative surgery significantly decreases the risk of endometriosis recurrence and its related symptoms [11]. However, hormone-blocking drugs cause severe adverse effects, such as postmenstrual symptoms, and off-target effects in other hormone-responsive organs, including bone and the brain, in endometriosis patients [4,12]. Therefore, alternative endometriosis treatments to replace hormonal therapy are in high demand.

Endometriosis is an estrogen-dependent proinflammatory disease [1,7,13]. Although the exact causes of endometriosis remain unknown, the theory of retrograde menstruation, which is an efflux of menstrual blood and cells via the fallopian tubes, is the well-accepted hypothesis by which endometriosis develops and progresses [1]. However, while 90% of reproductive-aged women experience retrograde menstruation, only 10% of them are diagnosed with endometriosis [14]. Therefore, in addition to retrograde menstruation, other factors are likely involved in the pathogenesis of endometriosis. Although the exact etiology of endometriosis has not yet been elucidated, the heritability of endometriosis has been estimated at 50 percent. In addition, a few twin studies have shown that the risk of endometriosis is increased in monozygotic twins compared with dizygotic twins, which indicates the possible contribution of specific genes or genetic alternations to endometriosis [15,16]. Meta-analyses of genome-wide association studies have identified single nucleotide polymorphisms (SNPs) associated with endometriosis risk. For example, SNPs in 11 different loci located in or near *IL1A* (interleukin 1 alpha), *ETAA* (ETAA1 activator of ATR kinase), *RND3* (Rho family GTPase 3), *NFE2L3* (nuclear factor, erythroid 2 like 3), *WNT4* (Wnt family member 4), *ID4* (inhibitor of DNA binding 4), *CDKN2B-AS1* (cyclin-dependent kinase inhibitor 2B antisense RNA), *VEZT* (vezatin, adherens junctions transmembrane protein), *GREB1* (growth regulating estrogen receptor binding 1), and *FN1* (fibronectin 1) are correlated with the risk of endometriosis [17]. Among them, *IL1A* and *NFE2L3* are the genes associated with inflammatory pathways [17]. In addition, eight loci in the *IL1A* gene are related to the risk of endometriosis in Japanese population, which supports the involvement of immune and inflammatory responses in the development of endometriosis [18,19]. While those genes have not yet been validated as potential targets for endometriosis treatment, a recent study has identified and validated a gene named *Neuropeptide S receptor 1 (NPSR1)* as a potential target through DNA sequencing [20]. This study showed that deleterious low-frequency coding variants in *NPSR1* are overrepresented in patients with familial endometriosis, especially in moderate/severe stages. NPSR1 is strongly expressed in glandular epithelial cells in eutopic and ectopic endometrium, while its ligand NPS is mostly found in the stroma. Inhibition of NPSR1 with the small molecular inhibitor SHA 68R prevents the release of proinflammatory Tumor Necrosis Factor (TNF)α by monocytes in vitro. In addition, the administration of SHA 68R relieves inflammation and pain in mouse models of endometriosis. NPSR1 expression is increased in several inflammatory diseases, such as inflammatory bowel disease and asthma [21,22]. In addition, NPSR1 is found in macrophages and T lymphocytes [23]. Sundman et al. show that stimulating monocytic THP-1 cells with proinflammatory cytokines TNFα and Interferon (IFN)γ significantly increases NPSR1 isoform expression, suggesting that NPSR1 may be a key factor of proinflammatory cytokine signaling [24]. 

Genetic mutations in the exon-coding region of genes involved in cell adhesion and chromatin-remodeling complexes associated with endometriosis progression were identified [25]. There is no direct evidence showing how retrograde menstrual debris found outside the uterus is cleared in healthy women. However, the defective cytotoxic activity of Natural Killer (NK) cells and decreased phagocytic macrophage activity in endometriosis patients indirectly support the hypothesis that the impaired immunes surveillance system attributes to the pathogenesis of endometriosis [26,27]. In the peritoneal cavity of endometriosis patients, the immune cells are recruited and secrete excessive levels of proinflammatory cytokines (interleukin (IL)-1, IL-6, TNF, and IFNγ), which promote disease development and progression [1,28,29]. Additionally, dysregulation of growth factors is involved in endometriosis. For example, the serum levels of vascular endothelial growth factor (VEGF), fibroblast growth factor (FGF), hepatocyte growth factor (HGF), insulin-like growth factor-I (IGF-I), granulocyte-macrophage colony-stimulating factor (GM-CSF), epidermal growth factor (EGF), and platelet-derived growth factor (PDGF) are higher in women with advanced endometriosis than in women without endometriosis [30,31,32,33]. In addition to genetic mutations, alterations in immune and growth factor signaling pathways are also causal factors for endometriosis progression.

During retrograde menstruation, refluxed apoptotic endometrial cell debris in the peritoneal area is recognized by dendritic cells. Dendritic cells presenting endometrial autoantigens express increased type I IFNs to promote monocyte differentiation, dendritic cell survival, and cytotoxic T-cell activity, which may attack endometrial cells that present autoantigens in healthy women [34,35]. However, type 1 IFN-mediated cell death signaling is dysregulated in endometriotic cells in endometriosis patients, and thus, these cells can evade the host immunosurveillance system. Type II IFN (IFNγ) fails to induce apoptosis in ectopic endometrial stromal cells, especially ovarian endometriotic cyst stromal cells, unlike in eutopic endometrial stromal cells with endometriosis and normal endometrial stromal cells [36]. While the underlying mechanisms of the IL family and TNF in endometriosis have been well studied, the IFN signaling pathways in endometriosis are not fully understood [1,35,37]. IFN signaling also plays a significant role in normal endometrium, especially during pregnancy [38].

This review summarizes the current knowledge of IFN signaling in endometriosis and in normal human endometrium. In addition, we discuss the functional interactions between estrogen and IFN signaling in the pathogenesis of endometriosis. Finally, clinical studies that examine the potential of IFN as a novel nonhormonal treatment of endometriosis are discussed.

## 2. IFN Signaling

IFNs are classified into types I, II, and III based on the type of receptors they bind. IFNs are secreted by host cells in response to viral infections or antigen recognition. In addition, IFNs function in antiviral activities via their pro- and anti-inflammatory effects.

### 2.1. Types of IFNs

Type I IFNs include interferon-alpha (IFNα), interferon-beta (IFNβ), interferon-epsilon (IFNε) in some placental mammals, and interferon-delta (IFNδ) and interferon-tau (IFNτ) in non-primate and non-rodent placental mammals [39]. In general, IFNα and IFNβ are expressed in leukocytes, fibroblasts, and endothelial cells in response to viral infections. They are expressed in the early phase of viral infections and induce the innate immune response by suppressing viral replication. Type I IFNs bind to a specific receptor complex on the cell surface known as the IFNα receptor (IFNAR), which consists of two chains, IFNAR1 and IFNAR2. The only type II IFN is interferon-gamma (IFNγ) produced by T cells and NK cells. IFNγ binds to a two-chained cell surface receptor complex that consists of IFNGR1 and IFNGR2 [40]. Type III IFNs, including interferon lambda (IFNλ), function similarly to type I IFNs, except that they only bind to the IL-10Rβ/IL-28R receptor complex and are selectively expressed on epithelial mucosal surfaces to immune-modulate adaptive immune responses to viral infection [39,41,42].

### 2.2. IFN Signaling

All three types of IFNs can activate the Janus-activated kinase-signal transducer and activator of transcription (JAK-STAT) pathway by binding to their receptors (Figure 1). In the case of type I and type III IFNs, the receptors activate JAK1 and tyrosine kinase 2 (TYK2), which results in tyrosine phosphorylation of STAT1 and STAT2 [43]. Phosphorylated STAT1 and STAT2 dimerize and interact with IFN regulatory factor 9 (IRF9) to form an IFN-stimulated gene factor 3 (ISGF3) complex. The ISGF3 complex then translocates to the nucleus, recognizes IFN-stimulated response elements (ISREs) in promoter regions, and induces gene transcription. Unlike type I and III IFNs, engagement of IFNγ with its receptors activates STAT1-mediated signaling through STAT1 phosphorylation by JAK1 and JAK2 without STAT2 activation. The phosphorylated STAT1 homodimers then translocate to the nucleus, where they bind to the IFNγ-activated site (GAS) in the promoters of interferon-stimulated genes (ISGs) for transcriptional regulation. In addition to the STAT-mediated canonical pathways, IFNs can noncanonically activate mitogen-activated protein kinase (MAPK) pathways and phosphatidylinositol-3-kinase/mammalian target of rapamycin (PI3K/mTOR) pathways to modulate cellular pathways [44].

## 3. Roles of Interferons in Human Endometrial Function

### 3.1. Type I IFNs

The endometrium is the innermost lining of the uterus and changes during the menstrual cycle in response to ovarian hormones. The endometrium consists of columnar epithelial cells, the underlying stroma containing numerous embedded glands, immune cells, and vascular endothelial cells. This lining becomes thickened during the proliferative phase (i.e., before ovulation) and prepares for a possible pregnancy during the secretory phase (i.e., after ovulation). The thickened endometrium is shed during menstruation when progesterone and estrogen levels fall [45]. In conjunction with ovarian hormones, IFNs modulate pregnancy [38,46]. 

Type I IFNs are highly expressed in the preimplantation blastocyst, trophoblast, and decidua trophectoderm during decidualization in mice [47]. In addition, preimplantation embryos secrete IFNα in vitro [48]. Therefore, type I IFNs might be crucial in embryo implantation and decidualization progression (Figure 2). 

IFNα stimulation synergically induces the expression of IFN-regulated gene 1 (*IRG1*) in humans at peri-implantation along with E_2_, although the underlying mechanisms are still unknown [49]. The transcription factor IFN regulatory factor 1 (IRF1) promotes implantation and is expressed in the human endometrium throughout the menstrual cycle and during peri-implantation in response to type I IFN exposure [50,51]. The levels of *IFNAR1* and *IFNAR2* are increased in the menstrual stage than in the proliferative phase in human endometrium [52]. IFNAR1 and IFNAR2 expressions are mostly found in the glandular epithelial cells but not in endometrial stromal cells, suggesting that glandular cells are the main cell type mediating type I IFN signaling in human endometrium [52]. In addition, ISGs, including interferon regulatory factor 8 (*IRF8*), interferon α-responsive protein *(IARP)*, interferon-activated gene 202B *(IFI202b)*, interferon γ-inducible GTPase 1 *(IIGP1*), interferon-stimulated gene 12 *(**ISG12)*, and *ISG15*, are upregulated in decidual tissues to enhance the decidualization of endometrial stromal cells in mice (Figure 2) [53].

Stimulation of human uterine epithelial cells with poly (I:C), a viral double-stranded RNA (dsRNA) mimic, increases IFNβ and ISG expressions [54]. In addition, inhibition of IFNβ signaling either with IFNβ-neutralizing antibody or IFNAR2-blocking antibody partially prevents the increases of ISG expressions, including MxA, OAS2, and PKR upon poly (I:C) stimulation, indicating that epithelial IFNβ exerts antiviral responses (Figure 2) [54]. 

While IFNε expression is widely found in the female reproductive tract, its expression in the endometrium is hormonally regulated during the menstrual cycle [55]. Unlike the vagina and ectocervix, in which IFNε is expressed in the basal and parabasal layers of the epithelium, IFNε is also detected in the surface of the endometrial luminal epithelium, suggesting that IFNε may play a role in immune protection from infection in the endometrium [55]. In female mice, IFNε-gene-deleted mice (IFNε^−/−^) are more susceptible to Herpes Simplex Virus (HSV)-2 and *Chlamydia muridarum* bacteria than wildtype mice [56]. 

In addition, vaginal concentrations of IFNε are lower in genital HSV-infected pregnant women than in healthy pregnant women [57]. The amniotic fluid concentration of IFNε is elevated in women undergoing spontaneous preterm labor with intra-amniotic infection than in women with no infection or sterile intra-amniotic inflammation [58]. The IFNε expression level is significantly elevated in the endometrium during the luteal phase than during the follicular phase. Progesterone receptor (PR) and its ligand progesterone regulate IFNε expression, in which progesterone stimulation interferes with the activation of IFNε promoter in endometrial epithelial cells in vitro and ex vivo [55]. In contrast, uterine IFNε expression is increased in estrogen-treated ovariectomized mice, which suggests hormonal regulation in IFNε expression [56]. The excessive expression of IFNε in the endometrium and its change during the menstrual cycle suggests that IFNε plays an essential role during pregnancy [55]. The vaginal concentration of IFNε is increased during pregnancy [57]. IFNε is gradually expressed in the myometrium from mid- to late gestation [58]. Therefore, IFNε exerts antiviral and anti-bacterial effects in the endometrium and during pregnancy (Figure 2).

### 3.2. Type II IFNs

IFNγ, the only type II IFN, is highly expressed in human trophoblast cells in the first trimester, while its receptor IFNGR is found in the placenta throughout the pregnancy [59]. Decidualized human endometrial stromal cells treated with conditioned media from human trophoblasts upregulate IFN-related genes, including *IFNGR1* and *JAK2* [60]. IFNγ is expressed in luminal and glandular uterine epithelial cells during the estrus stage in mice [61]. Additionally, IFNγ expression is highly increased within implantation sites in mice, which suggests the possible involvement of IFNγ in the implantation process [62]. Human uterine NK cells secrete cytokines, including IFNγ, and IFNGR is found in endometrial epithelial cells [63,64]. Ablation of NK cells, IFNγ, IFNGR, or STAT1 in mice causes defects in the decidual remodeling of spiral arteries that supply blood to the placenta. Moreover, the administration of IFNγ to NK-cell knockout (KO) mice led to the recovery of decidual artery remodeling [65]. Therefore, IFNγ may play a critical role in decidualization during the implantation process in humans. 

### 3.3. Type III IFNs

In addition to type I and II IFNs, type III IFNs also modulate normal endometrial functions. For example, IFNλ1 has an essential role in the immune defense of the placenta against viral pathogens, as human uterine epithelial cells and fibroblasts secrete IFNλ1 after exposure to the synthetic dsRNA viral ligand poly (I:C) [66]. Sex hormones also modulate IFNλ1 signaling because estrogen suppresses, but progesterone stimulates, IFNλ1-induced ISG expression in uterine epithelial cells [66]. Therefore, IFNλ1 might differentially modulate endometrial function based on menstrual cycle stage, pregnancy, and menopausal status. 

## 4. Dysregulation of Interferon Signaling in Endometriosis

Although the roles of IFNs in endometriosis have not been extensively studied, the currently available studies indicate that IFN signaling is disrupted in ectopic lesions and in eutopic endometrium of endometriosis patients, which may contribute to the pathogenesis and symptoms of the disease [35,67]

### 4.1. Dysregulation of IFN Signaling in Endometriotic Tissues

Type I IFN signaling pathways have been shown to be elevated in endometriosis. The mRNA levels of IFNα and IFNAR2 are higher in the eutopic endometrium of endometriosis patients than in the endometrium of women without endometriosis [68]. JAK1, a key modulator of type I IFN signaling, is significantly upregulated in endometriotic lesions compared with the eutopic endometrium of endometriosis patients [69]. Therefore, an elevation of the IFN/IFNAR2/JAK axis is essential for endometriosis. Human IFNα-2b treatment reduced the area of experimental endometriosis in a rat model [70]. IFNα-2b and IFNβ-1a, human recombinant type I IFNs, inhibited the proliferation and migration activity of primary endometrial stromal cells isolated from women with deeply infiltrating endometriosis [67]. However, a study of fifty-two infertile patients with moderate or severe endometriosis showed that intraperitoneal administration of human recombinant IFNα-2b after endometriosis surgery recurred endometriosis after 21 months [71]. The median time to recurrence after surgery is about 30 months [6]. IFNα-2b treatment after surgery shortened the time to recurrence, therefore, IFNα-2b treatment is not suitable as an endometriosis treatment. However, the debatable observations regarding the effects of type 1 IFN on endometriosis progression may result from the fact that happenings in an isolated cellular system do not always translate to a multicellular organism. 

In addition to type I IFN, the level of type II IFN (IFNγ) is significantly elevated in the peritoneal fluid of endometriosis patients compared with fluid from healthy women [72,73]. The soluble intercellular adhesion molecule-1 (ICAM-1) level is elevated after IFNγ stimulation in cultured cells isolated from eutopic endometrium, ovarian endometrioma, and peritoneal endometriotic lesions [74]. These may indicate that increased IFNγ may enhance the adhesive and invasive activity of the ectopic lesions to enhance endometriosis progression [74]. In cell-based functional assays, IFNγ neither inhibits cell proliferation nor induces apoptosis in endometrial stromal cells isolated from ectopic lesions due to elevated expression of antiapoptotic protein, B-cell lymphoma 2 (Bcl-2) and B-cell lymphoma-extra large (Bcl-X_L_), which prevents IFNγ-induced apoptosis [36]. Together, these findings indicate that elevation in peritoneal fluid IFNγexpression during endometriosis may enhance lesion adhesion but no lesion apoptosis due to elevated antiapoptotic protein expression by ectopic cells [75]. 

In contrast, Wu et al. demonstrated that the peritoneal level of IFNγ is significantly decreased in endometriosis patients [73]. They also found that the peritoneal level of IFNγ is inversely correlated with the serum level of soluble intercellular adhesion molecule-1 (ICAM-1) in endometriosis patients [73]. Therefore, the role of IFNγ is not clearly described in the pathogenesis of endometriosis due to the contractual observations. 

### 4.2. Dysregulation of STAT3 in Endometriosis

STAT3 activation negatively regulates STAT1-mediated cellular processes (Figure 3) [76]. Activated STAT3 (phosphorylated STAT3) competes for binding with STAT1 and prevents the formation of the STAT1:STAT1 homodimer [76]. Therefore, IFNα/IL-6-activated STAT3 inhibits STAT1-dependent gene expression, such as the induction of proinflammatory cytokines, including CXCL9 and CXCL10 and the transcription factor IRF1, by sequestering activated STAT1 into STAT1:STAT3 heterodimers and reducing the DNA binding of STAT1:STAT1 homodimers [77]. Additionally, activated STAT3 interacts with and cooperates with phospholipid scramblase 2 (PLSCR2) to negatively regulate the IFN response [78]. In carcinomas, constitutive STAT3 activation also reduces the expression of ISGF3 components (such as IRF9, STAT1, and STAT2) and IRF7, a transcription factor that participates in IFN response-related gene expression by directly binding to gene promoters [79]. Therefore, activation of STAT3 directly reduces the STAT1-mediated IFN response. In addition to blocking the activation of STAT1, the activated STAT3 dimer binds GAS and then modulates gene expression profiles involved in cell invasion, proliferation, anti-apoptosis (cell survival), angiogenesis, and inflammation in renal cell carcinomas [80]. However, it is not clearly described whether STAT3 regulates STAT1 in endometriosis yet.

The levels of activated STAT3 were significantly elevated in proliferative and secretory phase endometriotic tissues from women with endometriosis compared with tissues from women without the disease [81]. In addition, the levels of activated STAT3 (PIAS3) protein inhibitor, which negatively regulates STAT3 activity, were significantly lowered in women with endometriosis [82]. Treatment of IFNγ in endometriotic epithelial cells suppresses PIAS3 expression in a dose-dependent manner, consequently increasing pSTAT3 expression [82]. A proinflammatory cytokine IL-6 is elevated in the peritoneal fluid and serum of women with endometriosis and induces prolonged activation of STAT3 via association with the IL-6 receptor to enhance endometriosis progression [83,84,85]. Since the IL-6/STAT3 axis is highly activated in endometriotic tissues compared with normal endometrium, JAK/STAT3 signaling inhibitors, such as curcumol and tofacitinib, effectively suppress endometriosis progression in rat and mouse endometriosis models, indicating STAT3 as a possible target for endometriosis treatment [86,87]. 

### 4.3. Role of IFNs in Mesenchymal Stem Cells (MSCs)

The retrograde menstruation theory has limitations in explaining every case of endometriosis: First, not all women develop endometriosis, while most women experience retrograde menstruation. Second, it is rare, but premenarcheal girls, female fetuses, and men experience endometriosis, suggesting the involvement of other mechanisms in the pathogenesis of endometriosis [88,89]. Third, based on the stem cell hypothesis of endometriosis progression, MSCs contribute to the formation of endometriotic lesions in endometriosis patients [90]. The molecular properties of MSCs involved in endometriosis progression differ from those of normal MSCs [91]. For example, compared with MSCs in women without endometriosis (normal MSCs), MSCs in endometriosis patients (endometriotic MSCs) express higher levels of Cyclin D1, Matrix metalloproteinases (MMP)-2, and MMP-9. Additionally, the BCL2-associated X (BAX)/BCL-2 ratio is significantly lower in endometriotic MSCs than in normal MSCs. The expression of inflammatory genes, such as *IL-1β*, *IL-6*, *IL-8*, and *NF-κB*, and that of stemness genes, including SRY-Box Transcription Factor 2 (*SOX2*) and Spalt Like Transcription Factor 4 (*SALL4*), is also highly elevated in endometriotic MSCs compared with normal MSCs. Furthermore, endometriotic MSCs express higher levels of VEGF than normal MSCs [92]. Ectopic MSCs exhibit more aggressive migration and invasion characteristics than eutopic MSCs in vivo and in vitro [91]. Endometriotic MSCs are more immunosuppressive and exhibit significantly higher expression of immunosuppressive molecules than normal MSCs [93]. For example, endometriotic MSCs induce more spindle-shaped macrophages, which express substantially higher levels of CD14 and CD163; high levels of these markers are features of M2 macrophages. This finding suggests that endometriotic MSCs promote immunosuppressive M2 macrophages that may support growth and reduce immunosurveillance of ectopic lesions [94,95].

MSC therapy is considered a novel therapy for endometriosis due to its anti-inflammatory and anti-proliferative effects. Due to the distinct differences in molecular properties between endometriotic and normal MSCs, the transfer of normal MSCs has been used to treat endometriosis in a rat model of endometriosis. The transfer of MSCs derived from normal adipose tissue leads to decreases in the size of ectopic endometrial lesions in rats with endometriosis due to the downregulation of proinflammatory cytokines, including IFNγ and TNFα, in peritoneal macrophages and ectopic endometrial lesions that are required for endometriosis progression [96]. In the murine model of endometriosis, intravenous administrated adipose tissue-derived MSC (Ad-MSC) directly migrates to endometriosis lesions and inhibits the growth of endometriosis lesions, by suppressing the expressions of proinflammatory cytokines, including monocyte chemotactic protein-1 (*MCP1*), interleukin-6 (*IL6*), and leukemia inhibitory factor (*LIF*) and pro-fibrotic cytokine such as tumor growth factor-β1 (*IGFB1*) [97]. However, the efficacy of MSC therapy for endometriosis is still controversial. Abomaray et al. insist that MSC therapy should not be considered endometriosis therapy because MSC from allogeneic adipose tissue promotes cell proliferation and reduces cell apoptosis of endometriotic stromal cells in vitro [98]. These contradictory results may result from the difference between rodent models and human cells. MSCs from ectopic endometrium produce more IFNγ than MSCs from eutopic endometrium in vitro [99]. Secretion of indoleamine 2,3-dioxygenase 1 (IDO1) is a biomarker for the immunomodulatory activity of bone marrow-derived MSCs [100]. Priming endometrial MSCs with a high concentration of IFNγ alone or in combination with TNFα increase the release of IDO1 [101]. In addition, MSC priming increases expressions of adhesion molecules such as NCAM-1 and ICAM-1, suggesting that cytokines IFNγ and TNFα synergically promote the immunomodulatory capacity of endometrial MSCs [101].

## 5. The Interplay between Estrogen and IFN Signaling in Endometriosis Progression

Endometriosis is an estrogen-dependent inflammatory disease [1,7,102]. Therefore, estrogen and the estrogen receptors (ERs) ERα and ERβ play essential roles in the development and progression of endometriosis, as the levels of both estrogen and estrogen receptors in endometriotic lesions are elevated. Compared with the ERα level, the ERβ expression level is significantly higher in ectopic endometrial tissues than in normal endometrium [102,103]. Additionally, ERβ directly binds to the promoter region of ERα and blocks its expression, which leads to a progesterone-resistant state in endometrial tissues [104]. Thus, the Estrogen/ERβ axis is critical in endometriosis progression. Analysis of the ERβ-regulated transcriptome and the ERβ-cistrome revealed that the Estrogen/ERβ axis directly downregulates IFNα and IFNγ signaling pathways in endometriotic lesions in a mouse model [105]. Endometriotic cells resist IFNγ-induced cell death signaling [36]. The Estrogen/ERβ axis suppresses IFNα and IFNγ signaling pathways in ectopic endometrial lesions to promote the progression of endometriosis [105]. Furthermore, ERβ also suppresses the IFNα and IFNγ signaling pathways in the eutopic endometrium of mice with endometriosis, and mice with endometrium-specific ERβ overexpression mice are infertile [105]. This observation implies that Estrogen/ERβ axis-induced dysregulation of IFN signaling in the endometrium is involved in endometriosis-associated infertility. It has not been elucidated in humans yet.

## 6. Is IFN a Potential Target for Nonhormonal Endometriosis Treatment?

The current treatment for endometriosis is primarily based on hormonal therapy due to the estrogen-dependent nature of the disease. Combined or progesterone-only oral contraceptives are the first-line treatment for endometriosis because exogenous progesterone prevents the production and secretion of estrogen by the ovaries [12]. In addition, gonadotropin-releasing hormone (GnRH) agonists have been applied for endometriosis treatment because they also decrease ovarian hormone production [106]. In addition, the aromatase inhibitor letrozole lowers systemic estrogen levels by blocking the enzyme aromatase, which converts androgens into estrogen [107]. These treatments efficiently relieve endometriosis-associated pain and prevent endometrial cell proliferation [1,4,12]. However, the requirement for long-term hormonal therapy treatment leads to postmenopausal symptoms, including hot flashes and night sweats, in endometriosis patients, primarily during their reproductive years [108]. In addition, these therapies also cause off-target effects in other hormone-responsive tissues, such as bone and the brain [1,12]. These limitations in current hormonal therapies for endometriosis have led to the development of nonhormonal treatment approaches.

Based upon several in vitro and in vivo studies, IFNs are a potential target for nonhormonal endometriosis treatment. Human recombinant type I IFNs, such as IFNα-2b and IFNβ-1a, was shown to significantly reduce the proliferation and migration of endometrial stromal cells derived from endometriotic lesions in vitro [67,109]. The inhibitory effects were more dramatic with IFNβ-1a, as IFNβ-1a inhibited cell migration by suppressing the ERK-MAPK pathway. In addition, a high dose of IFNβ-1a induced cell cycle arrest and apoptosis [67]. Further, the intraperitoneal and subcutaneous administration of IFNα-2b in a rat model of endometriosis significantly reduces endometrial ectopic lesion size [70]. In addition, long-term subcutaneous administration (15 doses every 48 hours) of IFNα-2b was more effective in reducing the size of endometriosis implants than short-term administration (3 doses every 48 hours) [110].

In the clinical study, stage-dependent administration of IFNα-2b decreased disease severity with reductions in lesion size, symptom relief, and improved pregnancy rate in women with endometriosis while not causing any apparent side effects [71,111]. In another randomized clinical trial, however, endometriosis patients administered IFNα-2b had a higher recurrence rate of endometriosis than those treated with a vehicle, indicating that IFNα-2b may be inappropriate due to the risk of earlier recurrence of symptoms [71]. Further research is therefore required to define whether IFN pathways might be a potential molecular therapeutic target for endometriosis treatment.

## 7. Discussion

IFN signaling is essential in the implantation process in normal endometrium, and dysregulation of IFN signaling is associated with endometriosis progression. During retrograde menstruation, refluxed endometrial fragments activate host immune surveillance systems and elevate proinflammatory cytokines (such as TNFα and IFNs) in the pelvic area. Proinflammatory cytokines activate cell death signaling in refluxed endometrial fragments to be eliminated from the pelvic area. In this context, TNFα-mediated apoptosis signaling is activated in removing refluxed endometrial fragments. IFNs can cause cell death signaling in refluxed endometrial fragments. For example, type I IFNs can cause necroptosis through the ISGF3 complex [112]. Therefore, the synergism of TNFα-induced apoptosis and IFN-mediated necroptosis might have a critical role in the prevention of endometriosis progression by effectively removing endometrial debris. The dysregulation of both pathways should play a crucial role in the immune evasion of refluxed endometrial fragments that develop into endometriotic lesions. However, the way in which IFN-mediated necroptosis is dysregulated in endometriotic lesions in the process of endometriosis progression has not been thoroughly investigated. Therefore, further studies on the molecular mechanism of the dysregulation of IFN-mediated necroptosis in endometriotic lesions are needed.

Cytokines are a double-edged sword. In addition to apoptosis, TNFα is involved in cell survival signaling. In endometriotic lesions, TNFα activates NF-κB and inflammasome signaling to enhance endometriosis progression [113,114]. In this context, infliximab, an anti-TNFα monoclonal antibody, has been applied to treat endometriosis and relieve endometriosis-associated pain. However, evidence of the clinical benefits of infliximab for endometriotic lesions, dysmenorrhea, dyspareunia, and pelvic tenderness is lacking [115,116]. This observation implies that other cytokine-mediated cell survival systems might be involved in endometriosis progression in addition to the TNFα-mediated cell survival process. In this context, IFN would be considered another potential cell survival factor in endometriotic lesions, as high levels of IFNs are observed in endometriosis patients compared with women without endometriosis. Targeting IFN might provide dual benefits to endometriosis patients because IFN signaling is dysregulated in both endometriotic lesions and in eutopic endometrium. Therefore, anti-IFN therapy causes regression of endometriotic lesions, rescues endometrial function in endometriosis patients, and improves pregnancy rates. However, how IFN signaling is dysregulated in endometriosis and eutopic endometrium has not been fully elucidated. Therefore, the molecular mechanism of the dysregulation of IFN signaling in endometriotic lesions should be further investigated.

In the cell survival process, TNFα signaling intersects with IFN signaling. For example, NF-κB activated by TNFα interacts with STAT3 activated by IFNα to enhance NF-kB target gene expression [117]. In this context, the combination of anti-TNFα and anti-IFN should be further investigated to determine whether this combination synergistically suppresses endometriosis progression and relieves endometriosis symptoms without the adverse effects associated with current hormonal therapy.

## Figures and Tables

**Figure 1 biomolecules-12-01554-f001:**
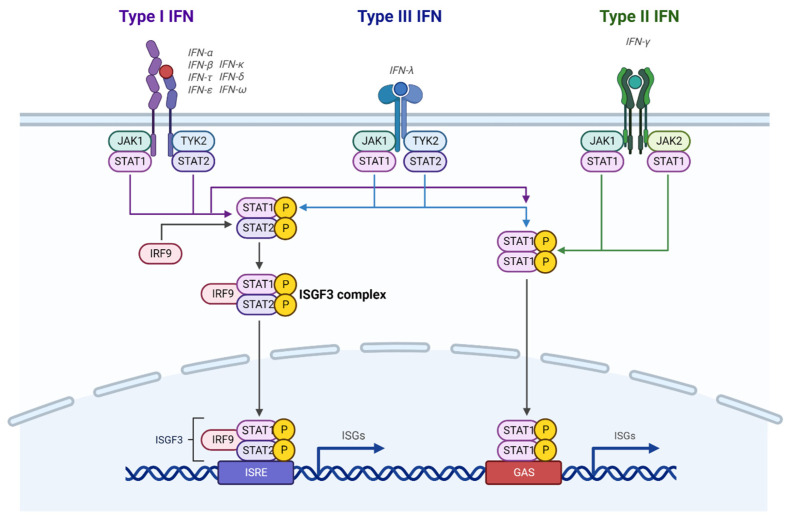
Canonical IFN signaling. Type I and Type III IFNs activate JAK1 and TYK2, which phosphorylates STAT1 and STAT2. Type II IFNs activate JAK1 and JAK2, which only phosphorylates STAT1. The activated STATs either form a complex with IRF9 or homodimer to stimulate ISG expression. This figure was created with BioRender.com.

**Figure 2 biomolecules-12-01554-f002:**
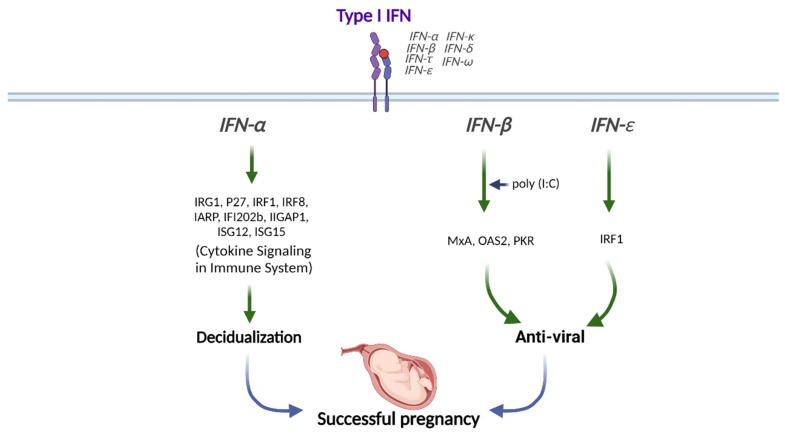
Role of Type I IFN in human pregnancy. Type I IFNs (IFNα, IFNβ, and IFNε) differentially modulate the downstream signals for successful pregnancy. This figure was created with BioRender.com.

**Figure 3 biomolecules-12-01554-f003:**
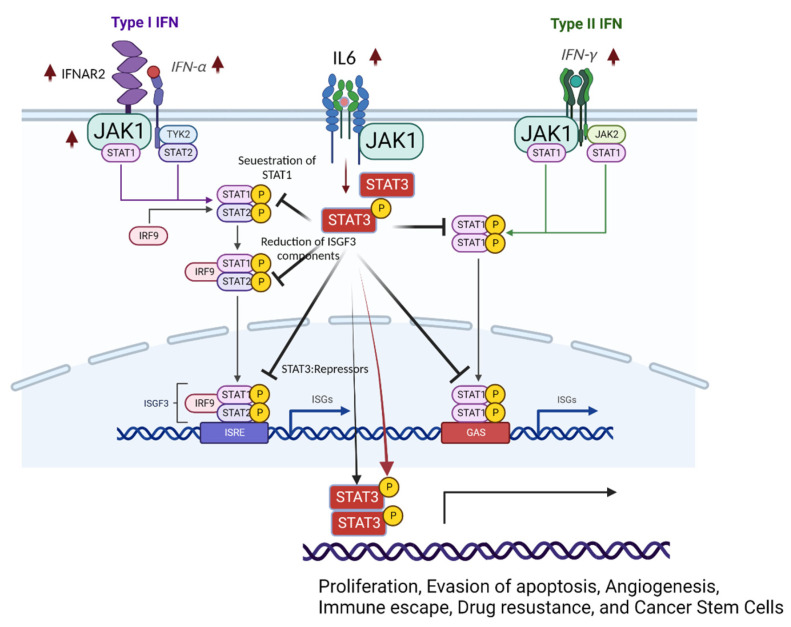
Dysregulation of IFN signaling in endometriotic lesions. Type 1 and Type II IFNs, JAK1, and IFNAR2 are elevated in endometriotic lesions compared with normal endometrium. In addition, the IL-6/STAT3 axis is activated in endometriotic lesions. Activated STAT3 negatively regulates STAT1 signaling in endometriotic lesions and increases the expression of STAT3 target genes to enhance endometriosis. This figure was created with BioRender.com.

## Data Availability

Further inquiries can be directed to the corresponding author.

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
