# Peer review of "Interferon Signaling in the Endometrium and in Endometriosis"

_biomolecules, 2022, doi:10.3390/biom12111554_

Round 1

Reviewer 1 Report (Previous Reviewer 1)

Thank you for making corrections and modifications to the previous submission. 

Reviewer 2 Report (Previous Reviewer 2)

Reviewer comments have been addressed sufficiently by authors.

This manuscript is a resubmission of an earlier submission. The following is a list of the peer review reports and author responses from that submission.

Round 1

Reviewer 1 Report

Please see attached word document. 

Author Response

Dear Editors,

We thank the reviewers for their generous comments on the manuscript.

Here, we have edited the manuscript to address their concerns.

We believe that the manuscript is now suitable for publication in Biomolecules.

Reviewer 1.

Comment: Major issues are within the description of IFN and their involvement in reproduction. The authors try to summarize information regarding IFN signaling during reproductive process across multiple species but do so disregarding mode or reproduction and implantation type. For example, the investigators focus on the function of IFN , a type I IFN secreted by the early cow embryo and indicate its function is in implantation, however, bovine embryos do not implant like the humans or rodent embryo and its major function is related to a process called maternal recognition of pregnancy. This miss understanding is highlighted by a figure (Figure 1) depicting

IFN  driving implantation of what appears to be a human embryo. IFN  is only produced by ruminant embryos which elongate (60 x 2 mm) before releasing IFN  during the critical period of maternal recognition of pregnancy and superficially implanting. A major issue with this section is that the investigators review information regarding IFN  from only a few generic publications and the information is highly superficial. For instance, they list only a few identified signaling mechanisms and genes identified in as few as four generic manuscripts. A quick search in PubMed reveals hundreds of papers covering this topic (many of which are recent reviews) with just as many IFN  induced genes identified in ruminant endometrium. Several reviews regarding the function of IFN  during early pregnancy have been published, specifically by authors Bazer, Roberts, Thatcher and Spencer, pioneers in the function of IFN  during early reproductive process in ruminants and other animals. None of these authors or their work was mentioned highlighting a lack of detail and depth in this section of the review.

Answer: We totally understand about the reviewer’s concern that we have not included much information about the role of IFN  in the review. It is true that Figure 1A could lead to the misunderstanding that pregnancy in ruminants is similar to that in women. Therefore, we modified figure 1A based on the reviewer’s comments.

We also explained the elongation process in lines 172-177. Maternal recognition of pregnancy increases the chance of pregnancy maintenance. Especially, IFN  is a key in maternal recognition of pregnancy in ruminants. Unlike in humans and mice, the bovine conceptus undergoes a morphological change called elongation after day 14 of pregnancy, in which the length of the conceptus dramatically increases, and the extraembryonic membrane starts to differentiate. The elongated conceptus secretes IFN  before implantation, inhibiting the luteolytic mechanism.

We also added another ISG in lines 199-203. Like other types I IFNs, IFN  also activates JAK/STAT signaling by binding to IFNAR1 and IFNAR2 and induces expression of ISGs such as MX1 and ISG15 in the uterus. ISG15 is upregulated in the ovine corpus luteum after IFN  administration. Interfering ISG15 expression impairs the early bovine embryo development with lower hatching blastocyst.

We also added the molecular function of IFN  in early pregnancy studied by Bazer, Roberts, Thatcher, and Spencer (lines 146-166).

In healthy women, endometrial cells outside the uterus are monitored and cleared by the immune surveillance system.

Comment: Please include a citation. What do the author mean by the immune surveillance system?

Answer: There is no direct evidence showing how retrograde menstrual debris found outside the uterus is cleared in healthy women. However, the defective cytotoxic activity of NK cells and decreased phagocytic macrophage activity in endometriosis patients indirectly support the hypothesis that the impaired immune surveillance system contributes to the pathogenesis of endometriosis (Lines 70-74)

Line 119-132.

Comment: This section needs citations.

Answer: We cited the section.

The endometrium consists of columnar epithelial cells and the underlying stroma, which contains numerous embedded glands.

Comment: The endometrium also contains various immune cells and vascular endothelial cells.

Answer: We revised the statement. The endometrium consists of columnar epithelial cells, the underlying stroma containing numerous embedded glands, immune cells, and vascular endothelial cells (Lines 136-138).

Comment: Is this during pregnancy or during the cycle? This should be in the context of pregnancy as IFNT is released from the conceptus.

Answer: It is at the mid-stage of the estrous cycle (line 169)

Comment: What do the authors mean by surrounding membranes? IFNT is secreted by mononuclear trophoblast cells in cattle and sheep. I’m surprised this sentence does not include citations or mention of maternal recognition of pregnancy.

Answer: We revised the line and the section based on the comment and highlighted it in pink.

Maternal recognition of pregnancy increases the chance of pregnancy maintenance. Especially, IFN  is a key in maternal recognition of pregnancy in ruminants. Unlike in humans and mice, the bovine conceptus undergoes a morphological change called elongation after day 14 of pregnancy, in which the length of the conceptus dramatically increases, and the extraembryonic membrane starts to differentiate. The elongated conceptus secretes IFN  before implantation, which consequently inhibits the luteolytic mechanism (lines 172-177).

Comment: IFNT inhibits estrogen receptor expression in endometrium of sheep but not cattle.

Answer: we revised it. The released IFNτ inhibits the expression of estrogen receptor (ER) and oxytocin receptor (OTR) in the luminal and glandular epithelium of sheep, which consequently blocks endometrial PGF production during early pregnancy, allowing for successful implantation (Lines 182-185).

Comment: This sentence is not clear to me.

Answer: Embryonic IFNτ activates embryonic IFNτ receptor 1 and endometrial IFNτ receptor 2, which directs normal uterine preparation for early bovine implantation (Lines 185-187).

Comment: What type of endometrial cell? Please include a citation.

Answer: YAP, a Hippo pathway modulator, translocates to the nucleus in the endometrial epithelial cells of cattle and mice during early pregnancy to modulate the extracellular matrix of the endometrium (Lines 187-189).

Comment: This sentence is taken directly from the end of the abstract of this publication. This study is of blood neutrophils and is not directly related to endometrial immune cells.

Answer: We reparaphrased the sentence. Even though it is not directly related to endometrial immune cells, this study helps to understand how IFN  regulates successful pregnancy in cows.

In vitro, NET formation is impaired by IFNτ in blood neutrophils collected from heifers via JAK3/PI3K signaling pathways, which may lead to successful implantation (Lines 194-196).

Comment: This section does not seem cohesive. Also, the authors may consider reviewing the actions of IFNdelta during early pregnancy in swine and horses.

Answer: We added a paragraph. IFN  is expressed by porcine trophoblastocyst during early pregnancy. Unlike IFN , IFN  does not have antiluteolytic activity in female pigs. Instead, IFN  in pigs and horses exerts antiviral activity by inhibiting the cytopathic effect of Vesicular Stomatitis Virus (VSV) in vitro. IFN  is also found in the horse conceptus and pig conceptus. In addition, IFN  in the pig conceptus activates IRF1 in the uterine stroma (lines 204-208).

Figure 1.

Comment: IFNT is only expressed by the ruminant conceptus and particularly during conceptus elongation. This figure includes an image of a spherical blastocyst with trophoblast

expansion/implantation like humans. Further, the figure legend does not effectively describe.

Answer: We revised the figure and added, “IFNT is only expressed by the ruminant conceptus and particularly during conceptus elongation.”

Comment: Please revise (ISG12)12 to “ISG12”

Answer: We revised it. Interferon-stimulated genes (ISGs), including interferon regulatory factor 8 (IRF8), idiopathic acute recurrent pancreatitis (IARP), interferon-activated gene 202B (IFI202b), interferon-inducible GTPase 1 (IIGP1), interferon-stimulated gene 12 (ISG12), and ISG15, are upregulated in decidual tissues to enhance the decidualization of endometrial stromal cells (Line 233)

Comment: Please include a citation.

Answer: We added a citation. The pig conceptuses undergoing implantation secrete IFNϒ to recruit various subpopulations of T cells, such as proliferating cell nuclear antigen (PCNA)-positive T cells and CD4+ T cells.

Comment: IFNAR2 was previously abbreviated. Please keep abbreviations consistent throughout.

Answer: We revised it. The mRNA levels of IFNα and IFNAR2 are higher in the endometrium of endometriosis patients than in the endometrium of women without endometriosis (Lines 275-277)

Comment: JAK1 was previously abbreviated.

Answer: We revised the sentence. Moreover, JAK1, a key modulator of type I IFN signaling, is significantly upregulated in endometriotic lesions compared with the eutopic endometrium of endometriosis patients (Lines 277-279)

Comment: What is IFNalpha 2b and IFNbeta 1a? I may have missed this, but was information related to these two IFNT provided before this line?

Answer: IFNalpha 2b and IFNbeta 1a are human recombinant Type I IFNs. There are 13 subtypes for IFNalpha and 1 subtype for IFNbeta in humans. The subtypes are highly conserved but exert different functions (Gibbert 2013). We revised the sentence. IFNα-2b and IFNβ-1a, which are human recombinant type I IFNs inhibited the proliferation and migration activity of primary endometrial stromal cells isolated from women with deeply infiltrating endometriosis (Lines 282-285)

Comment: Please be consistent when describing IFNalpha 2b.

Answer: We revised it to IFNα-2b (Line 280).

Comment: IRF previously abbreviated.

Answer: We revised it (line 315).

Comment: Can the author’s please elaborate on the effects of IFNG on MSC in this section?

Answer: We added a paragraph and highlighted it in blue. MSCs from ectopic endometrium produce more IFN  than MSCs from eutopic endometrium in vitro. Secretion of indoleamine 2,3-dioxygenase 1 (IDO1) is considered a biomarker for the immunomodulatory activity of MSCs. Priming endometrial MSCs with a high concentration of IFN  (100 ng/ml) increases the release of IDO1. The combination of TNF and IFN  further increases the release of IDO1. In addition, the priming increases expressions of adhesion molecules such as NCAM-1 and ICAM-1, suggesting that cytokines IFN  and TNF  synergically promote the immunomodulatory capacity of endometrial MSCs (Lines 360-367)

Therefore, estrogen and the estrogen receptors (ERs) ERα and ERβ play essential roles in the development and progression of endometriosis, as local levels of both estrogen and estrogen receptors are elevated.

Comment: Local where?

Answer: The E2 and ER levels are increased in endometriotic lesions. We specified the location based on the comment. ERα and ERβ play essential roles in the development and progression of endometriosis, as the levels of both estrogen and estrogen receptors in endometriotic lesions are elevated (Lines 370-373).

Comment: Please add citation.

Answer: We added the citation (Line 385).

Comment: Is IFNbeta 1 the same as IFNbeta 1a described above?

Answer: Yes, it is actually IFNbeta 1a. We revised it. The inhibitory effects are more dramatic with IFNβ-1a, as IFNβ-1a inhibits cell migration partially by suppressing the ERK-MAPK pathway. In addition, a high dose of IFNβ-1a decreases the percentage of cells in G2/M phase and induces late apoptosis (Line 410)

Line 421-424.

Comment:  Should this be IFN not INF?

Answer: We revised it (Line 441 and 443).

Comment: NFKB was previously abbreviated.

Answer: We revised it (Line 446).

Reviewer 2 Report

Please see my comments on the attached file.

Author Response

Dear Editors,

We thank the reviewers for their generous comments on the manuscript.

Here, we have edited the manuscript to address their concerns.

We believe that the manuscript is now suitable for publication in Biomolecules.

Reviewer 2.

Comment: Please fix this statement.

It is my understanding that 90% of women who had a laparoscopy whilst actively menstruating exhibited retrograde menstruation into the pelvic cavity IRRESPECTIVE of endometriosis diagnosis. So it is presumed that 90% of all menstruators exhibit retrograde but only 10% go on to develop endometriosis.

Answer: While 90% of reproductive-aged women experience retrograde menstruation, only 10% of them are diagnosed with endometriosis (Lines 43-44).

Comment: Whilst this is an exciting study, how is it linked to the scope of the review? Are there any links between NPSR1 and interferon regulation?

Answer: We propose IFN signaling as a nonhormonal target for endometriosis in this review. Here, NPSR1 has been validated as a nonhormonal target of endometriosis. While only a few studies identified several genes as possible nonhormonal targets via meta-analyses, the NPSR1 inhibitor significantly reduces the inflammatory response in endometriosis. It is still unclear whether IFN also stimulates NPSR1 in endometriosis, but their relationship in other inflammatory conditions provides the possibility. In addition, it is helpful to add NPSR1 to understand possible nonhormonal targets for endometriosis. Therefore, we think that NPSR1 is relevant to the review. We also added a paragraph explaining that IFN  significantly increases NPSR1 isoform expression in monocytes and highlighted in yellow (Lines 63-67).

The added paragraph is that ‘NPSR1 expression is increased in several inflammatory diseases such as inflammatory bowel disease and asthma. In addition, NPSR1 is found in macrophages and T lymphocytes. Sundman et al. show that stimulating monocytic THP-1 cells with proinflammatory cytokines TNF  and IFN  significantly increases NPSR1 isoform expression, suggesting that NPSR1 may be a key factor of proinflammatory cytokine signaling (Lines 63-67).

Comment: Dendritic cells? Please amend throughout.

Answer: We amended dendric cells to dendritic cells on lines 85-89.

Comment: On all endometriotic cells or a specific cell type? Please provide more information if possible.

Answer: We specified cell types. Type 1 IFN-mediated cell death signaling is dysregulated in endometriotic cells in endometriosis patients, and thus, these cells can evade the host immunosurveillance system. IFN , type II IFN, fails to induce apoptosis in ectopic endometrial stromal cells, especially ovarian endometriotic cyst stromal cells, unlike in eutopic endometrial stromal cells with endometriosis and normal endometrial stromal cells (Line 89-93).

  1. Figure 1A.

Comment: IFNe mentioned in the figure but not mentioned in the text. Given recent publications on IFNe during the menstrual cycle and in pregnancy, it would be beneficial to the manuscript to add this information. 

Answer: We added a paragraph about the role of IFN  in endometrium and pregnancy (Lines 204-255).

  1. MSC therapy is considered a novel therapy for endometriosis due to its anti-inflammatory and antiproliferative effects.

Comment: In what sort of capacity? To neutralize retrograde menstrual fragments as they enter the pelvic cavity? Or to actively stop the growth of ectopic lesions?

Given that MSCs and epithelial stem cells are found in lesions, how would an MSC therapy work?

Answer: MSC therapy is considered a novel therapy for endometriosis due to its anti-inflammatory and anti-proliferative effects. In the murine model, intravenous administrated adipose tissue-derived MSC (Ad-MSC) directly head to endometriosis lesions and inhibit the growth of endometriosis lesions by suppressing the expressions of proinflammatory cytokines including monocyte chemotactic protein-1 (Mcp1), interleukin-6 (Il6), and leukemia inhibitory factor (Lif) and pro-fibrotic cytokine such as tumor growth factor-β1 (Tgfb1). However, the efficacy of MSC therapy for endometriosis is still controversial. Abomaray et al. insist that MSC therapy should not be considered endometriosis therapy because Ad-MSC promotes cell proliferation and reduces cell apoptosis of endometriotic stromal cells in vitro (lines 352-367).

Round 2

Reviewer 1 Report

This area of research is extremely important and this review is justified. Although they seem to be made at minimal, thank you for making corrections to the manuscript.

Author Response

Thanks